# Thermal and Calorimetric Investigations of Some Phosphorus-Modified Chain Growth Polymers 2: Polystyrene

**DOI:** 10.3390/polym14081520

**Published:** 2022-04-08

**Authors:** Malavika Arun, Stephen Bigger, Maurice Guerrieri, Paul Joseph, Svetlana Tretsiakova-McNally

**Affiliations:** 1Institute of Sustainable Industries and Liveable Cities, Victoria University, P.O. Box 14428, Melbourne, VIC 8001, Australia; stephen.bigger@vu.edu.au (S.B.); maurice.guerrieri@vu.edu.au (M.G.); 2Belfast School of Architecture and the Built Environment, Ulster University, Newtownabbey BT37 0QB, UK; s.tretsiakova-mcnally@ulster.ac.uk

**Keywords:** polystyrene, phosphorus-containing compounds, additive and reactive routes, thermal stability, combustion attributes

## Abstract

In this paper, we report on the thermal degradation behaviours and combustion attributes of some polymers based on polystyrene (PSt). Here, both additive and reactive strategies were employed, through the bulk polymerization route, where the modifying groups incorporated P-atom in various chemical environments. These included oxidation states of III or V, and the loading of phosphorus was kept at ca. 2 wt.% in all cases. The characterization techniques that were employed for the recovered products included spectroscopic, thermal, and calorimetric. It was found that the presence of different modifying groups influenced the degradation characteristics of the base polymer, and also exerted varying degrees of combustion inhibition. In all cases, the modification of the base matrix resulted in a noticeable degree of fire retardance as compared to that of the virgin material. Therefore, some of the modifications presented have the potential to be explored on a commercial scale.

## 1. Introduction

Polystyrene (PSt) is a relatively inexpensive, readily available, and transparent thermoplastic polymer that is used for a wide number of applications [1]. The monomer, styrene, may also be copolymerised with other monomers to produce co- and/or ter-polymers, often having improved properties [2]. Some of the modified systems can be utilized for specific functions, such as removing heavy metals from water [3,4,5,6]. Some of the multicomponent systems include styrene acrylonitrile (SAN), acrylonitrile-butadiene-styrene (ABS) polymers, styrene-butadiene rubber (SBR), etc. Furthermore, there are two types of polystyrene foams that are commercially available: expanded polystyrene (EPS) and extruded polystyrene (XPS). Expanded polystyrene is generally used for food packaging applications and XPS, which is a higher density foam, is used in the building sector [7]. One of the disadvantages of PSt, despite its useful characteristics, is its relatively high flammability. When ignited, polystyrene and its copolymers often burn quickly with a visible flame, releasing volatiles including styrene monomer, oligomers, lower hydrocarbons such as benzene, lower alkylbenzenes [8], etc. During the burning process, polystyrene can also melt, flow, and drip, which can lead to an increased fuel load feeding into enhanced flame spread [9]. Generally, combustion of unmodified polystyrene produces a minimal amount of char residue.

Polystyrene homopolymer generally starts degrading at a temperature around 270 °C and continues until 425 °C under normal conditions in air. Through random main-chain cleavages and associated processes, PSt forms varying amounts of a number of compounds. These include styrene, benzaldehyde, styrene oxide, acetophenone, and l-phenylethanol. Such scissions generally originate from structural irregularities, such as head-to-head linkages and other minor structures, and are predominant at relatively lower temperatures [10]. Styrene and benzaldehyde are found to be the prominent fractions among the decomposition products [11]. Like the majority of main-chain carbon polymers, it has long been established that the thermal decomposition of polystyrene generally occurs in three steps: initiation, propagation, and termination, which follows a radical chain mechanism [12]. Most of the low molecular weight fragments that are produced upon the thermal or thermo-oxidative degradation of styrenic polymers can form combustible mixtures with ambient air. These mixtures, in the presence of a suitable ignition source, can subsequently undergo flaming combustion.

With a view to address the flammability issue of polystyrene-based materials, several flame retarding strategies have been developed. In this context, for adequately fireproofing the finished products, both the additive and reactive approaches are often adopted [13,14,15,16,17,18,19]. Given the environmental implications and the need to use relatively higher loadings, the use of halogen-based fire retardants (FRs) and heavy metal-based compounds or their combinations are progressively being discouraged. In this context, FRs based on phosphorus have gained prominence, especially in recent years [20,21,22]. For achieving an acceptable level of fire retardance, formulation based on bio-sourced FRs are also used [23,24,25]. The relative predominance of condensed- and vapour-phase activities of phosphorus-based FRs depend both on the nature of the P-containing additive or reactive (i.e., chemical class of the compound and the oxidation state of the P-atom), as well as on the chemical constitution of parent polymeric matrix [26,27,28,29,30,31].

Both additive and reactive strategies have been employed to bring about noticeable improvements in the flame retardance of polystyrene at a relatively low loading of phosphorus (ca. 2.0 wt.% in all cases). Here, additives were simply incorporated into the monomer or initiator mixture to form physical mixtures, whereas the polymerizable compounds were added into the main monomer mixture during the syntheses of copolymers. All polymerization reactions were conducted through the bulk route, with minor adaptations if necessary, as previously reported [32,33]. The novelty of the present work stems from the utilization of a number of phosphorus-bearing compounds or groups, where the P-atom exhibited its valence states of III or V, and that the chemical environments of these groups also belonged to different categories (i.e., phosphites, phosphates, phosphonates, phosphorylaminoester, phosphine, and phosphine oxide). The central idea here was to gauge the influences, through a systematic approach, of the oxidation states and chemical environments of the modifying groups on the combustion properties of polystyrene. Hence, the loading of P (i.e., ca. 2 wt.%) was kept constant in all cases.

## 2. Materials and Methods

### 2.1. Materials

Except for the additives 9,10-dihydro-9-oxa-10-phosphaphenanthrene-10-oxide (DOPO) and diethyl-1-propylphosphonate (Thermofisher Scientific, Melbourne, Australia), all other chemicals and solvents were purchased from the Aldrich Chemical Company, Melbourne, Australia. The solid compounds were used as received, whereas liquid reagents and solvents were optionally dried by keeping them over molecular sieves (4 Å). The thermally labile initiators and monomers were stored under sub-ambient temperatures in a refrigerator, or in a freezer, as required. A proprietary inhibitor column was employed to remove the inhibitor, *tert*-butylcatechol, from the styrene monomer prior to use. The necessary precursors, comonomers, and the additive, diethylbenzylphosphonate (DEBP), were synthesized by following procedures as given in Part I (i.e., pertaining to polymethyl methacrylate) of our previous work [34]. The other additives and monomers studied included triphenylphosphine (TPP), triphenylphosphineoxide (TPPO), diethylphosphite (DEHPi), triethylphosphite (TEPi), triethylphosphate (TEPa), diethylpropylphosphonate (DEPP), diethyl-1-(acryloyloxyethyl)phosphonate (DE-1-AEP), acrylicacid-2-[(diethoxyphosphoryl)methylamino] ester (ADEPMAE), diethyl-2-(acryloyloxy)ethylphosphate (DEAEPa), and diethyl-p-vinylbenzylphosphonate (DEpVBP). The chemical structures of both the additives and reactives are also given in our previous work [34].

### 2.2. A Typical Procedure for Bulk Polymerization

The procedure was based on a previously published work where minor adjustments to the curing regime were affected [33]. Here, the required amount of monomer(s) and initiators were stirred thoroughly in a conical flask under a nitrogen atmosphere for ca. 5 h at 70 °C, until a visible increase in the viscosity was observed. The calculated amount of the additive or reactive was then added and stirred for another 1 h. The resultant mixture was subsequently poured into an aluminium pan of ca. 50 mL volume and the pan was stoppered with an aluminium lid. The pan was placed in an air oven preheated to 40 °C and kept for curing for about 20 h. In the second stage of curing, the temperature of the oven was raised to 60 °C for 8 h. After another 20 h of curing at 80 °C, the contents of the pan were allowed to cure again for a period of 3 h at 100 °C before being cooled to room temperature. The final plaque was then extracted from the aluminium pan. Again, a fixed phosphorus loading of 2 wt.% was used. The structures of the additives and comonomers used in the present study are given previously [34]. The required amounts of the modifying compounds relative to the monomer and associated details are given in Table 1.

### 2.3. Characterization

The structures and purities of the additives, precursors, monomers and polymeric materials were established primarily through NMR spectroscopy. For this, a Bruker 600 MHz instrument was employed, and the spectra were run in deuterated solvents (CDCl_3_, or _d6-_DMSO) at ambient probe conditions. The signals were calibrated against residual proton signal of the solvent, or phosphoric acid as the external calibrant, as required. The raw data were processed by using proprietary software from the manufacturer (TopSpin 4.0.8).

Thermo-gravimetric (TGA) analyses on the polymeric products were run in nitrogen at 10 °C min^−1^ and 60 °C min^−1^, from 30 to 900 °C, using a Mettler-Toledo instrument. The set heating rate of 60 °C min^−1^ was chosen with a view to compare and correlate the results from the TGA experiments to those of other calorimetric techniques, such as the pyrolysis combustion flow calorimetry (PCFC) technique. Differential scanning calorimetry (DSC) runs were primarily used to estimate the heats of pyrolysis of the various polymeric materials. For this purpose, the thermograms were recorded in a nitrogen atmosphere at a heating rate of 10 °C min^−1^, from 30 to 550 °C, using a Mettler-Toledo instrument. The glass transition temperatures of the samples were also obtained from the DSC curves.

The combustion behaviours of the various polymeric products were measured through pyrolysis combustion flow calorimetry (PCFC) technique, also known as ‘micro combustion calorimetry’ (MCC). This is a small-scale calorimetric testing method increasingly being used to analyse the fire behaviour of various solid materials when subjected to a forced non-flaming combustion, under anaerobic or aerobic conditions. The details of this technique, including working principle and information regarding the parametric outputs, are published elsewhere [35,36,37,38]. In the present work, PCFC runs were carried out using a FAA Micro Calorimeter at 1 °C min^−1^. The samples were pyrolyzed in an inert atmosphere with nitrogen at a temperature range of 100–750 °C and averages over triplicate runs are reported. The heats of combustion of various polymeric materials were deduced by performing a ‘complete’ combustion in pure oxygen, using an IKA C200 ‘Bomb’ calorimeter (IKA, Oxford, UK). Pelleted samples, weighing ca. 0.5 g, were placed inside a ‘bomb’ cell. The instrument was previously calibrated using recrystallized benzoic acid and, for each sample, triplicate runs were performed.

The software-based analyses of the data obtained through thermogravimetry were based on an algorithm and the accompanying software noted that was previously reported [39,40,41]. The detailed kinetic analyses, including the mathematical treatment of the data, are also given in the above reports. In the present work, thermograms obtained at a relatively low heating rate of 10 °C min^−1^ were chosen for the analysis, as this is expected to capture most of the underlying steps in the thermal degradative pathway of the material in question. Essentially, the process involved the initial transfer of the raw data from the TGA instrument as an Excel file followed by processing, where the main step(s) of degradation of the sample were initially identified.

## 3. Results and Discussion

Generally, acrylic-based polymers are used as transparent plaques for various applications, where favourable optical clarity and enhanced weather resistance are the main prerequisites. In this context, the rigid solid materials casted from PSt-based materials are not sufficiently explored to date. However, for such applications, the relatively high flammability of PSt can become a limiting factor. Hence, in the present study, plaques of PSt were prepared (ca. 50 g scale) through the incorporation of various additives and reactives by adopting some procedures that were previously reported [11,33]. It is also relevant to note that the loading of phosphorus, in all cases, was normalized to 2 wt.%, while altering the chemical environments and oxidation states of the phosphorus atom in the admixtures. Furthermore, both the additive and reactive routes were utilized, with a view to identify the influences, if present, between the two strategies on the combustion features of the polymeric products. In addition, given the relatively nominal loading of phosphorus (2 wt.%), any marginal improvements in the fire retardance of the modified systems as compared to the virgin polymers would be most advantageous. The products obtained through the bulk polymerization route were chosen for further and detailed investigations in terms of their thermal (TGA) and calorimetric (DSC) properties, as well as their combustion (PCFC and ‘bomb’ calorimetry) characteristics. As previously reported, the final compositions of the products through the reactive strategy were effectively controlled with a high degree of certainty since the polymerization reactions were driven to near completion, i.e., ca. 99% conversion; the latter was established through ^1^H NMR spectra of the obtained plaques [34].

The bulk polymerization route employed in the present study for making PSt-based materials was found to be successful. In almost all cases, dense and tough polymeric plaques were formed. However, some exceptions were noted. The DOPO-modified version of PSt was found to be substantially brittle, and was also prone to shattering quite easily under a mechanical stress. Furthermore, the reactively modified version with the P- and N-containing comonomer, ADEPMAE, was found to have a plasticizing effect on the final product. The results obtained through the thermal and calorimetric evaluation of the products prepared through the bulk polymerization method are given, in detail, in the following sections.

### 3.1. Thermogravimetric Analysis (TGA)

In this section, the relevant TGA parameters obtained for each PSt-based sample at a heating rate of 10 °C min^−1^ are presented in Table 2, and the corresponding thermograms are given in Figure 1, Figure 2, Figure 3, Figure 4, Figure 5 and Figure 6, respectively. It is relevant to note here that, generally, the thermal degradation of PSt is strongly temperature-dependent [17]. For instance, at modest temperatures, initial cleavage occurs at a head-to-head linkage in the main chain, followed by smooth extrusion of styrene monomer, which, in fact, is the basis for recycling of the polymer. The TGA data obtained at 60 °C min^−1^ were primarily used for comparison with the relevant parameters obtained through the PCFC runs. The thermograms obtained for PSt samples generally exhibited only one main degradation step. However, during the initial phases of degradation, some small mass losses can be also observed in almost all the cases. Generally, the induction temperatures for the modified systems were noticeably lower than that of the unmodified version (Table 2). Furthermore, this effect was significantly higher in certain systems compared to others. On the other hand, the temperature at 50 wt.% was higher in the case of some of the modified systems than polystyrene.

As mentioned before, the induction temperatures seem to be lowered, albeit to varying degrees, in the modified systems as compared to the parent polymer. This could be either due to the release of the additives or early thermal cracking of the pendent P-containing groups prior to the onset of the main-chain decomposition of the polymer. Furthermore, the temperature at 50 wt.% mass loss were higher in several systems (DOPO, DEHPi, TEPi, TEPa, DEBP, and DEpVBP), indicating an enhanced resistance to mass loss as compared to unmodified PSt. However, in all other systems a reverse trend was observed, except in the case of the system incorporating TPP, where the corresponding temperature remained unaltered. The systems with the additive DEHPi and reactive P–N monomer (ADEPMAE), produced the maximum amounts of char (6.9 and 6.7 wt.%, respectively). Other components have also assisted in producing more char (except TPP), although to different extents.

### 3.2. Kinetic Analysis of the TGA Thermograms

The apparent values of the energy of activation (*E_a_*) for PSt-based samples were obtained from the software runs [40]. It is relevant to note here that values quoted are averaged over the main decomposition step and the range of *α* values were also appropriately chosen. The results are collated in Table 3 below.

The virgin polymer was found to follow the kinetic model that was in conformance with first order (F1) kinetics (*E_a_* value typically in the region of 180 to 220 kJ mol^−1^). Therefore, this particular model was subsequently applied in analysing all the modified versions of the styrene polymers. As in the case of PMMA samples [34], a reduction in the average activation energy values can be observed, except for the system containing the phosphite additive (DEHPi); here a relatively higher value (305 kJ mol^−1^) was obtained. Therefore, it is to be inferred here that, except in the case of DEHPi, in all other cases, the modification resulted in decreased thermal stability of the parent polymer.

### 3.3. Differential Scanning Calorimetry

Heats of pyrolysis (∆*H_pyro_*) data and the values of the glass transition temperatures for various PSt-based bulk samples were obtained from the DSC runs of the samples at a heating rate of 10 °C min^−1^ in the temperature range of 30–550 °C. Given below are the results obtained for the PSt-based bulk samples from the DSC runs as well as their corresponding ∆*H_pyro_* and *Tg* values (Table 4).

Here, a gradual decrease in the values of ∆*H_pyro_* in the case of the modified materials can be observed, with the PSt+DEAEPa system exhibiting the lowest value of 324 mJ mg^−1^. Here, it can be assumed that in these systems, some degree of cooperative interaction between the polymeric matrix and the modifying groups are present during the phase changes that occur during a DSC run. It is also relevant to note that the modification with the phosphate comonomer DEAEPa showed the lowest values in both the systems. Furthermore, the modifications, in all cases, seem to lower the glass transition temperature as compared to the parent polymer matrix, which can be attributed to their varying plasticizing effects.

### 3.4. Pyrolysis Combustion Flow Calorimetry (PCFC)

The PCFC data of the PSt-based samples are shown below in Table 5.

From the above table it can be seen that all the modified versions have lower values of the relevant parameters, such as pHRR, THR, HRC, and EHC, when compared to the unmodified polystyrene sample. The material modified with the P–N monomer, ADEPMAE, shows the lowest values for pHRR (425 W g^−1^), THR (23.8 kJ g^−1^), HRC (424 J g^−1^ K^−1^), and EHC (25.2 kJ g^−1^), indicating that the modifying group has the most favourable effect amongst the systems under consideration [38,42]. It is also relevant to note that the char yields obtained for the different samples showed a wide variability, where no underlying trend can be found.

In addition, the first derivative of the TGA thermograms and HRR curves showed significant correspondence. Furthermore, all the polymeric systems showed a decreasing trend in the values of all the relevant parameters from the PCFC runs. This points towards the fact that when the modified versions of the PSt-based systems degrade, there appears to be a noticeable degree of cooperative interaction between the modifying groups and parent polymeric chains.

### 3.5. ‘Bomb’ Calorimetry

The values of the heat of combustion, Δ*H_comb_* obtained through the ‘bomb’ calorimetric runs are collected in Table 6.

All the modified samples showed a definite decrease in the ∆*H_comb_* values as compared to the unmodified version, which clearly demonstrated the vapour-phase inhibitory effect(s) of the modifying groups. Such an effect was found to be particularly pronounced in the case of PSt modified with ADEPMAE, the P–N-containing monomer, where it may be also assumed that there is some degree of P–N interaction, thus exhibiting the lowest value of ∆*H_comb_*, 33.03 kJ g^−1^ [42]. It is also highly relevant to note here that, generally, there may be some sort of interaction (physical or chemical) of the parent polymer matrix with the modifying groups also evident during pyrolysis and combustion (from DSC and PCFC tests, respectively).

### 3.6. Some Generalizations among the Test Parameters

Some generalizations were found to exist amongst some of the relevant test parameters from TGA, PCFC, and ‘bomb’ calorimetry techniques. However, no particular correspondences were observed among the values of the char yields from the two techniques. However, several systems (such as unmodified PSt and modified versions with TPP, DEHPi, TEPi, DE-1-AEP, ADEPMAE, DEAEPa, and DEpVBP) showed substantial char residues (ranging from 3 to 8 wt.%) in the PCFC runs, whereas for the others the corresponding values closely resembled the char yield from the TGA runs (≤1 wt.%). Furthermore, several of the modifying additives and groups seem to exhibit some degree of condensed-phase activity during the PCFC runs. This was also evident in terms of enhanced char yields produced by the modified systems as compared to virgin PSt.

The relative values of ∆*H_comb_* and EHC did not show any particularly discernible trend and were varied amongst the samples. However, PSt containing the P–N monomer (ADEPMAE) exhibited significantly lower values, and in fact, its ∆*H_comb_* EHC values are particularly lower as compared to all other systems [42]. This, again, confirms the significant combustion inhibitory effect of the monomer as is also evident in the relevant parameters obtained through TGA, PCFC, and ‘bomb’ calorimetric techniques [42].

## 4. Conclusions

Through the present investigation on the thermal properties and combustion attributes of PSt-based polymers that were mainly obtained through TGA and calorimetric measurements (DSC, PCFC, and ‘bomb’ calorimetry), some inferences can be drawn: (1) TGA: Generally, the presence of additives or reactives in the parent polymer matrix was found to enhance its thermal stability. However, in the case of those systems containing the additives, there were no clear indications of the additives influencing the mode of degradation of the base polymeric matrix. It is relevant to note here that the degradation of the PSt base matrix is strongly influenced by the heating rate [43]; (2) DSC: Here, a gradual decrease in the values of ∆*H_pyro_* in the case of all modified materials can be observed, with the PSt+DEAEPa system showing the lowest value. This can be attributed to the relative ease of volatilization and pyrolysis of the additive or reactive moieties as compared to the main-chain scission reaction of the parent polymeric matrix. Furthermore, the glass transition temperatures of the modified polymeric systems were noticeably lower than that of the unmodified counterpart. This can be attributed to the plasticizing effect of the additive compound and reactive group; (3) PCFC: All the modified versions showed lower values for the relevant parameters, such as pHRR, THR, HRC, and EHC, when compared to the unmodified polystyrene sample. The material modified with the P–N-monomer, ADEPMAE, showed the lowest values. Thus, these results point towards varying degrees of combustion inhibition in all the modified systems, presumably acting in the vapour phase; (4) ‘Bomb’ calorimetry: Here, the modified PSt-based samples showed a definite decrease in the ∆*H_comb_* values as compared to the unmodified version, which, again, clearly demonstrated the gaseous phase inhibitory effect(s) of the modifying groups. Such an effect was found to be particularly pronounced in the case of PSt modified with ADEPMAE, the P–N-containing monomer, where it can be also assumed that there is some degree of P–N interaction. The utility of the in-house developed software, in deducing the Arrhenius parameters from the TGA runs, was also explored and successfully applied, considering only a thermogram obtained through a single heating rate. The extended validity of these values should be treated with caution, and that such values can only be treated as ‘apparent’ values at best. The mode of action pertaining to P- and P–N-containing groups on the combustion attributes of the parent polymer matrix has been recently published [44].

## Figures and Tables

**Figure 1 polymers-14-01520-f001:**
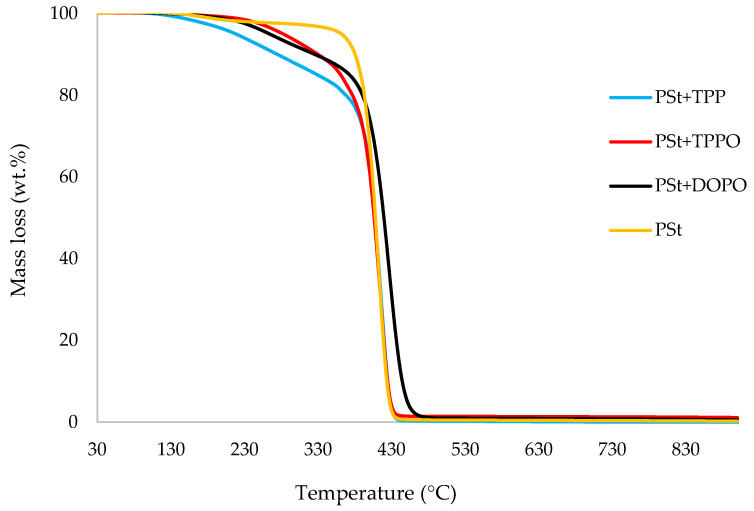
An overlay of the TGA curves of the PSt-based materials with solid additives, at 10 °C min^−1^ in nitrogen, from 30 to 900 °C.

**Figure 2 polymers-14-01520-f002:**
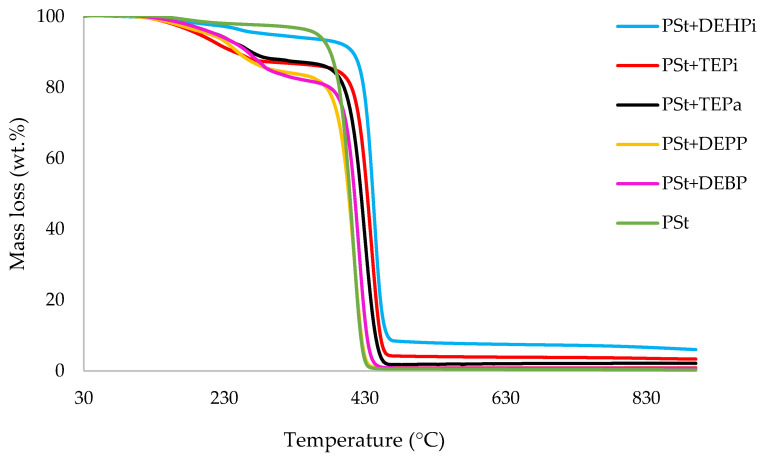
An overlay of the TGA curves of the PSt-based materials with liquid additives, at 10 °C min^−^^1^ in nitrogen, from 30 to 900 °C.

**Figure 3 polymers-14-01520-f003:**
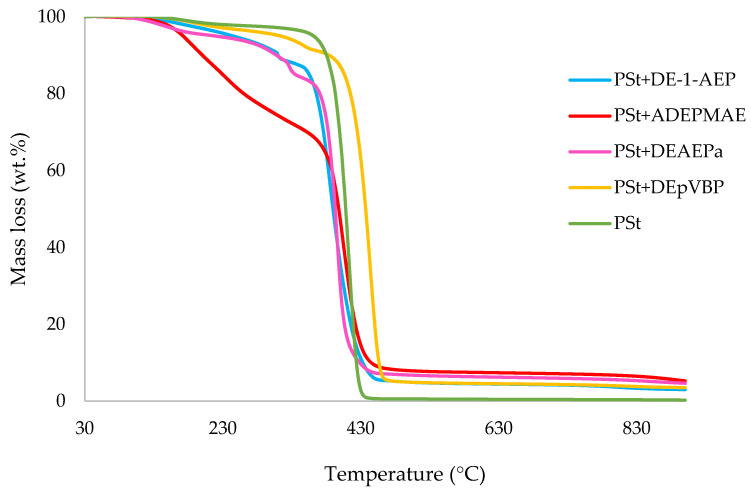
An overlay of TGA curves of the PSt-based materials with reactives, at 10 °C min^−^^1^ in nitrogen, from 30 to 900 °C.

**Figure 4 polymers-14-01520-f004:**
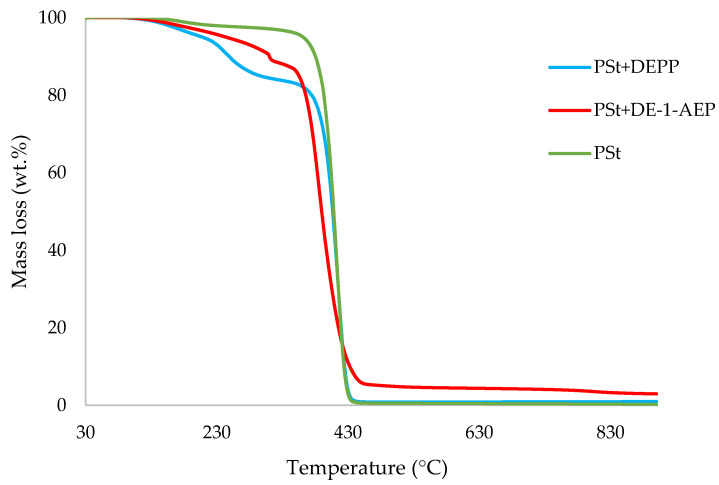
An overlay of the TGA curves of PSt and PSt+aliphatic phosphonate materials, at 10 °C min^−1^ in nitrogen, from 30 to 900 °C.

**Figure 5 polymers-14-01520-f005:**
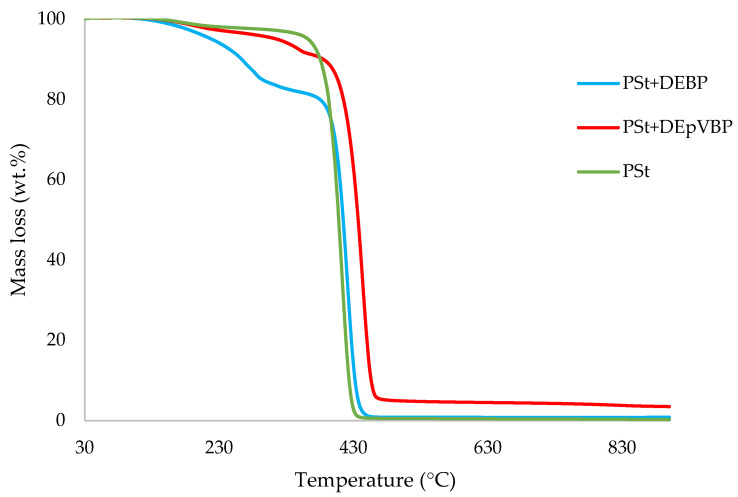
An overlay of the TGA curves of PSt and PSt+aromatic phosphonate materials, at 10 °C min^−1^ in nitrogen, from 30 to 900 °C.

**Figure 6 polymers-14-01520-f006:**
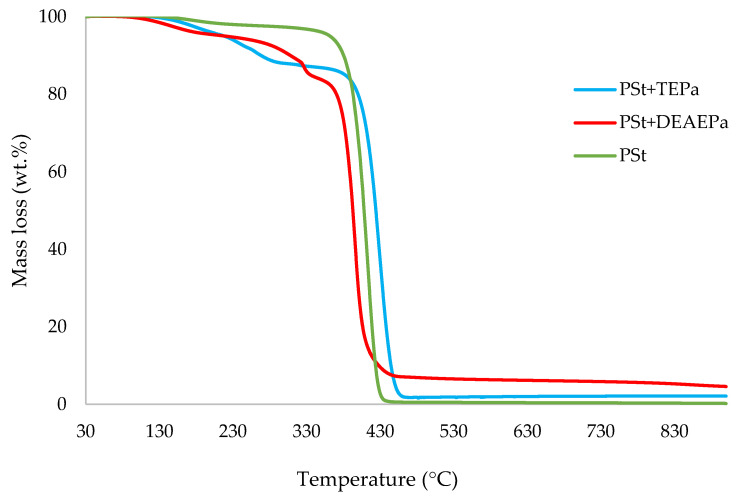
An overlay of the TGA curves of PSt and PSt+phosphate materials, at 10 °C min^−^^1^ in nitrogen, from 30 to 900 °C.

**Table 1 polymers-14-01520-t001:** Details of the preparative data for St-based bulk polymers *.

Sl. No.	Styrene (mL)	Additive/Reactive	Formula Weight	Additive/Reactive Weight (g/mL)	BPO/Dicumyl Peroxide (mg)
1	50.00	---	---	---	50.0/25.0
2	36.57	TPP	262	6.76 g	44.0/22.0
3	36.10	TPPO	278	7.17 g	43.0/21.5
4	37.90	DOPO	216	5.57 g	43.0/21.5
5	40.09	DEHPi	138	3.56 mL	44.0/22.0
6	39.29	TEPi	166	4.42 mL	44.0/22.0
7	38.84	TEPa	182	4.40 mL	44.0/22.0
8	39.00	DEPP	180	4.65 g	44.0/22.0
9	37.53	DEBP	228	5.88 g	44.0/22.0
10	23.32	DE-1-AEP	236	3.81 g	26.0/13.0
11	23.00	ADPMAE	265	4.27 g	26.0/13.0
12	23.00	DEAEPa	252	4.07 g	26.0/13.0
13	23.00	DEpVBP	254	4.10 g	26.0/13.0

* since the pans were tightly stoppered during the curing phase, a near quantitative yield was obtained in each case (~99 wt.%).

**Table 2 polymers-14-01520-t002:** Some relevant parameters from the TGA analyses carried out in nitrogen at a heating rate of 10 °C min^−1^ of PSt-based systems.

Sl. No.	Sample	Induction Temp. (°C)	Temp. at 50 wt.% (°C)	Residue at 500 °C (wt.%)	Final Residue at 800 °C (wt.%)
1	Polystyrene	129	408	0.5	0.4
2	PSt+TPP	105	408	0.2	0.0
3	PSt+TPPO	123	407	1.4	1.3
4	PSt+DOPO	118	420	1.0	0.8
5	PSt+DEHPi	73.0	441	8.1	6.9
6	PSt+TEPi	75.0	433	4.1	3.6
7	PSt+TEPa	119	424	1.8	2.1
8	PSt+DEPP	96.0	407	0.9	0.9
9	PSt+DEBP	100	415	0.9	0.8
10	PSt+DE-1-AEP	109	390	4.9	3.6
11	PSt+ADEPMAE	72.0	398	7.9	6.7
12	PSt+DEAEPa	74.0	393	6.8	5.5
13	PSt+DEpVBP	121	437	4.9	4.0

**Table 3 polymers-14-01520-t003:** Energy of activation (*E_a_*) and other relevant parameters of PSt-based samples obtained using the software (model: F1 First Order).

Sl. No.	Sample	Apparent Activation Energy (*E_a_*, kJ mol^−1^)	A (min^−1^)	*R*^2^ Values	*α*-Value Range
1	PSt	270	2.68 × 10^20^	0.998	0.1 to 0.9
2	PSt+TPP	118	2.69 × 10^8^	0.9125	0.2 to 0.8
3	PSt+TPPO	162	9.52 × 10^11^	0.9706	0.2 to 0.8
4	PSt+DOPO	165	8.94 × 10^11^	0.9939	0.2 to 0.8
5	PSt+DEHPi	305	1.23 × 10^22^	0.9943	0.1 to 0.9
6	PSt+TEPi	229	3.88 × 10^16^	0.9908	0.2 to 0.8
7	PSt+TEPa	210	2.24 × 10^15^	0.9935	0.2 to 0.8
8	PSt+DEPP	157	3.82 × 10^11^	0.9652	0.2 to 0.8
9	PSt+DEBP	137	6.79 × 10^9^	0.9218	0.2 to 0.8
10	PSt+DE-1-AEP	97	1.00 × 10^7^	0.9815	0.1 to 0.9
11	PSt+ADEPMAE	130	3.82 × 10^9^	0.9978	0.4 to 0.9
12	PSt+DEAEPa	216	4.26 × 10^16^	0.9871	0.2 to 0.8
13	PSt+DEpVBP	223	1.08 × 10^16^	0.9923	0.2 to 0.8

**Table 4 polymers-14-01520-t004:** Heats of pyrolysis data of PSt-based materials obtained from DSC tests.

Sl. No.	Samples	Heats of Pyrolysis, ∆*H_pyro_* (mJ mg^−1^)	Tg (°C) (±5 °C)
1	PSt	810	96
2	PSt+TPP	650	55
3	PSt+TPPO	720	76
4	PSt+DOPO	730	61
5	PSt+DEHPi	540	70
6	PSt+TEPi	560	65
7	PSt+TEPa	680	62
8	PSt+DEPP	690	74
9	PSt+DEBP	660	62
10	PSt+DE-1-AEP	560	63
11	PSt+ADEPMAE	470	70
12	PSt+DEAEPa	324	65
13	PSt+DEpVBP	520	75

**Table 5 polymers-14-01520-t005:** PCFC data of PSt-based materials.

Sl. No.	Samples	Temp to pHRR (°C)	pHRR (W g^−1^)	THR (kJ g^−1^)	HRC (J g^−1^ K^−1^)	Char Yield (wt.%)	EHC (kJ g^−1^)
1	PSt	434	840	37.1	852	4.4	38.8
2	PSt+TPP	434	682	35.9	686	6.5	38.4
3	PSt+TPPO	434	717	36.7	729	0	36.7
4	PSt+DOPO	446	618	36.3	621	0	36.3
5	PSt+DEHPi	462	778	32.9	778	5.9	35
6	PSt+TEPi	463	771	31.9	776	5.6	33.8
7	PSt+TEPa	444	743	34.2	772	0	34.2
8	PSt+DEPP	438	815	34.2	813	0	34.2
9	PSt+DEBP	440	792	34.5	794	0	34.5
10	PSt+DE-1-AEP	409	501	33	499	7.4	35.6
11	PSt+ADEPMAE	431	425	23.8	424	5.7	25.2
12	PSt+DEAEPa	416	757	31.9	755	6.4	34.1
13	PSt+DEpVBP	468	655	34.6	653	2.9	35.6

**Table 6 polymers-14-01520-t006:** Heats of combustion data for PSt-based samples from ‘bomb’ calorimetric measurements.

Sl. No.	Sample	Δ*H_comb_* (kJ g^−1^)
1	PSt	41.50
2	PSt+TPP	41.02
3	PSt+TPPO	40.30
4	PSt+DOPO	39.50
5	PSt+DEHPi	38.70
6	PSt+TEPi	38.89
7	PSt+TEPa	38.99
8	PSt+DEPP	39.55
9	PSt+DEBP	39.13
10	PSt+DE-1-AEP	38.23
11	PSt+ADEPMAE	33.03
12	PSt+DEpVBP	39.47

The ∆*H_comb_* of PSt+DEAEPa could not be performed.

## Data Availability

Not applicable.

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
