# Peer review of "Thermal and Calorimetric Investigations of Some Phosphorus-Modified Chain Growth Polymers 2: Polystyrene"

_polymers, 2022, doi:10.3390/polym14081520_

Round 1
Reviewer 1 Report
This study deals with Thermal and calorimetric investigations of some phosphorus- 2 modified chain-growth polymers (Polystyrene). Several experimental controlling factors were investigated and evaluated in this work. I recommend the manuscript for publication after considering the following suggestions which their addressing will fit the manuscript for publication.
Comments are given below:
- The abstract should be rewritten to summarize the work; the abstract should state briefly the purpose of the research, the principal results, and major conclusions.
- The introduction should be clarified in terms of uniqueness and advantage what is the novelty of this work over the previous related work. Many long sentences should be refined.
- These polymers can be applied for heavy metal removal from water, please mention this in the introduction, you can use these references: Sep. Purif. Technol. 43 (2005) 43-48; React. Funct. Polym. 65 (2005) 267-275; Sep. Purif. Technol. 42 (2005) 111-116; J. Environ. Chem. Eng. 4 (2016) 3632-3645.
- How many times do sorption experiments conduct for each condition? It's better to include standard errors for reported values if they were measured repeatedly.
- Elaborate the discussion parts of the results, More profound discussions and comparisons with other published works are welcomed.
- Add experimental conditions to captions of each figure.
Author Response
See enclosed.

Reviewer 2 Report
The manuscript entitled Thermal and calorimetric investigations of some phosphorus-modified chain growth polymers 2: Polystyrene presented to me for review is interesting and I think can interest a wide group of scientists working under improving thermal stability properties of polystyrene. This polymer is widely used in many industrial applications, and it is highly recommended to improve its flammability stability. However, there are many concerns that should be addressed before publication:
The results presented in the manuscript are explained in comparison to results obtained for PMMA modified with the same additives / reactants. But these results are not published yet, so it is hard for the reviewer to compare these results - properties and conclusions which are presented. Moreover, the results should be explained for this particular system based on PS, with different types of additives. It should be explained how different structures of these additives can influence the properties of obtained products.
The authors wrote that induction temperatures of thermal decomposition of modified PS seemed to be lower than induction temperatures for pristine PS (TGA analysis). Because they are lower by dozen-several dozen degrees (as it is presented in table 2 and on TGA curves) they are not seemed to be lower, by they are lower or even much lower. So decomposition starts earlier. The temperature at 50 wt-% is a little higher for some PS modified samples, so thermal decomposition starts earlier but it goes slower. Moreover, thermal decomposition goes by several steps so DTG analysis should be done to better analysis of this process.
What is the influence of used modifiers on the properties of prepared samples? The authors performed DSC measurements but they didn’t show any thermograms. What is the Tg of the prepared samples? The structure of additive has an influence on temperatures of characteristic transitions of polymer? The authors should show DSC thermograms and try to explain differences (if they are).
The authors should point out which modified PS products are better and explain why.
One cannot resist the impression that the manuscript is read like a research report with a rather scarce explanation of the dependencies present.
Author Response
See enclosed.

Round 2
Reviewer 1 Report
Author has revised the manuscript according to reviewer comments and can be accepted as it is.
Reviewer 2 Report
The Authors took all concerns under consideration and the manuscript can be accepted in it present form.